# Cobalt-catalyzed aminoalkylative carbonylation of alkenes toward direct synthesis of γ-amino acid derivatives and peptides

Le-Cheng Wang[1,2], Yang Yuan[1], Youcan Zhang[1] & Xiao-Feng Wu ®[1,2,3] ✉

γ-Amino acids and peptides analogues are common constituents of building blocks for numerous biologically active molecules, pharmaceuticals, and natural products. In particular, γ-amino acids are providing with better metabolic stability than α-amino acids. Herein we report a multicomponent carbonylation technology that combines readily available amides, alkenes, and the feedstock gas carbon monoxide to build architecturally complex and functionally diverse γ-amino acid derivatives in a single step by the implementation of radical relay catalysis. This transformation can also be used as a late-stage functionalization strategy to deliver complex, advanced γ-amino acid products for pharmaceutical and other areas.

Amino acids and their derivatives are essential for life as key motifs of proteins and pharmaceutical agents, and biologically relevant natural products. Among them, γ-amino acids (GABA), which are four-carbon non-protein amino acids, are an indispensable constituent of free amino acids in most living organisms[1–6]. In contrast to ubiquitous α-amino acids, γ-amino acid derivatives are an up-and-coming drug carrier (biocompatibility, degradability, and multifunctionality)[7], and γ-amino acids can also be used to modify polypeptide drugs, such as hydrolysis resistance, half-life, pharmacokinetics, and physiological properties[8–10]. As drug carrier, these properties could help it to enhance the safety and effectiveness of drugs while improving their bioavailability. In addition, γ-amino acids can better extend amino acid residues and encoded amino acid peptides with more possibilities because of the present of three carbon atoms between the nitrogen and carbonyl groups[11]. γ-Amino acids, as inhibitory neurotransmitters in the mammalian central nervous system (CNS), are gradually being widely used in medicine and also chemical industry due to their various physiological functions (Fig. 1A)[12–14].

However, concerning their synthetic methodologies, compared with the well-established synthesis of α- and β-amino acids, general catalytic approaches to afford γ-amino acids are relatively underdeveloped. In addition, the number of commercially available γ-amino acids is mainly limited to simple and natural amino acids, while most other γ-amino acids require complex synthetic procedures. Currently, γ-amino acids' chemical synthetic strategies mainly rely on the

hydrolysis of the corresponding γ-lactam (or γ-butyrolactone) or the Michael-type addition of carbonyl and nitro compounds[2,9,15–19]. These traditional synthetic routes towards γ-amino acids are limited by multistep preparation and narrow substrates scope, which seriously reduces the efficiency of the synthesis process (Fig. 1B). Transition metal-catalyzed carbonylation represents one of the most effective and powerful strategies for the installation of carbonyl group[20–23]. However, carbonylation procedures to achieve the direct synthesis of four-carbon γ-amino acids have been elusive and challenging. Additionally, ethylene[24–27] and carbon monoxide are widely exploited in chemical industry with massive global production. The production capacity of ethylene, the simplest alkene, amounted to 201 million metric tons in 2020. Therefore, direct utilization of alkenes (especially ethylene) and carbon monoxide as starting materials will provide a more efficient, economical, and straightforward method for the preparation of γ-amino acid derivatives and peptides.

Attracted by the above discussed backgrounds, we proposed that an efficient and mechanistically distinct catalytic process involving simple abundant amines, alkenes, and carbon monoxide feedstock gas would provide a new strategy for the synthesis of amino acids. We envisioned that an α-aminoalkyl radical could serves as a highly reactive radical specie to start the transformation[28–30]. As a subsequent radical rarely reaction, the α-aminoalkyl radical could be captured by alkenes to form a new carbon radical. However, we found that the addition of this strong

[1]Dalian National Laboratory for Clean Energy, Dalian Institute of Chemical Physics, Chinese Academy of Sciences, Dalian, Liaoning, China. [2]Leibniz-Institut für Katalyse e.V., Rostock, Germany. [3]University of Chinese Academy of Sciences, Beijing, China. ✉e-mail: xwu2020@dicp.ac.cn

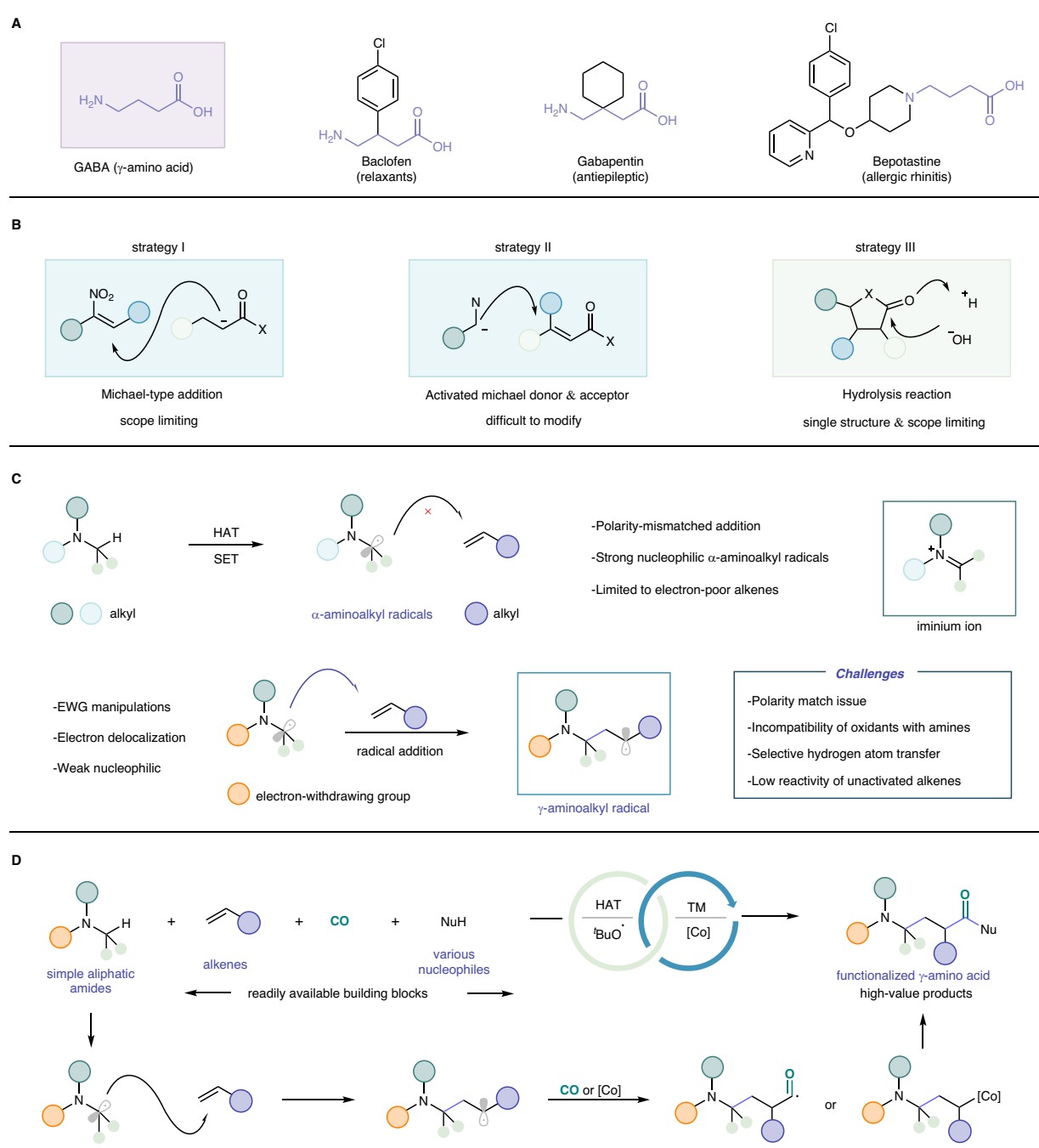

**Fig. 1 | Alkene aminoalkylcarbonylation and synthesis of γ-amino acid derivatives. A** Representative drugs demonstrating the ubiquity of γ-amino acid derivatives. **B** Chemical synthesis routes to γ-amino acids. **C** The challenges and conception of this work. **D** This work: cobalt-catalyzed aminoalkylcarbonylation of alkenes.

nucleophilic α-aminoalkyl radical to alkene has been limited to electron-poor alkenes and specific intramolecular examples due to mismatched polarity (Fig. 1C)[31–34]. The direct addition of α-aminoalkyl radicals to alkenes is thermodynamically unfavourable and readily radical-polar crossover further converted into undesired iminium ion in the presence of oxidant which needed for initiate the reaction[35–37]. Tuning the selectivity between α-aminoalkyl radical and iminium ion become the core for the success of this transformation. We hypothesised these challenges can be overcame as following: (1) by incorporating an electron-withdrawing group into the corresponding alkylamine to generate an electron delocalized α-aminoalkyl radical, which could decrease

its nucleophilicity and also suspend the radical-polar crossover; (2) we considered that a thermodynamically more favourable γ-aminoalkyl radical from alkene could facilitate carbonylative coupling by efficiently intercepting carbon monoxide. Herein, we report the successful accomplishment of this concept by incorporating an acyl group which could not only promote the addition of alkene but also delays the generation of iminium ions (Fig. 1D).

## Results
### Reaction development
We chose to begin with 4-phenyl-1-butene **2h** and amine **3a** as the model substrates to test the feasibility of this alkene aminoalkylative

**Table 1 | Condition optimization**

Reaction conditions: [a] **1a** (10 equiv.), **2a** (0.6 mmol), **3a** (0.3 mmol), Co(acac)$_2$ (5 mol%), (R, R)-$^t$Bu-Pybox (5 mol%), DTBP (3 equiv.), CO (40 bar), PhCF$_3$ (0.2 M), 120 °C, 20 h, yields were determined by GC-FID analysis. [b] Isolated yield. [c] 10 bar ethylene, 40 bar CO. [d] 10 bar ethylene and 20 bar CO. [e] 5 bar ethylene and 5 bar CO. *acac* acetylacetone, *hfacac* hexafluoroacetylacetone, *DTBP* di-*tert*-butyl peroxide.

| Entry | ⬤ | Deviation from above | Yield [%] |
|---|---|---|---|
| 1 | B | None | 79 (72)$^b$ |
| 2 | B | Co(hfacac)$_2$ instead of Co(acac)$_2$ | 25 |
| 3 | B | **L5** instead of (R, R)-$^t$Bu-Pybox | 75 (70)$^b$ |
| 4 | B | **L6** instead of (R, R)-$^t$Bu-Pybox | 45 |
| 5 | B | DCE instead of PhCF$_3$ | 32 |
| 6 | B | PhCl instead of PhCF$_3$ | 72 |
| 7 | B | 100 °C instead of 120 °C | 8 |
| 8 | B | 10 bar CO instead of 40 bar CO | 42 |
| 9$^c$ | B | ethylene instead of **2 h** | 83 (79)$^b$ |
| 10$^d$ | B | ethylene instead of **2 h** | 81 (78)$^b$ |
| 11$^e$ | B | ethylene instead of **2 h** | 72 |

carbonylation reaction. As shown in Table 1, we first identified a suitable class of electron-withdrawing group substituted amines which capable of undergoing the HAT process. We found that these substrates with modified groups could successfully occur in this reaction, such as acetyl, trifluoroacetyl, and aminoacyl. With dimethylacetamide (**1a**) as the substrate, we found that the combination of Co(acac)$_2$ and (R, R)-$^t$Bu-Pybox led to the best yield (79% GC yield and 72% isolated yield). The use of Co(hfacac)$_2$ as a catalyst lowered the yield of the targeted transformation. After optimizing a series of ligands, we found that terpyridine ligands gave the best reaction efficiency (Table 1, entries 3–4). Compared with PhCF$_3$ and PhCl, a decreased yield was obtained when this reaction was carried out in DCE (Table 1, entries 5–6). When the reaction temperature was reduced to 100 °C, the targeted reaction could hardly occur due to the low conversion of substrate (Table 1, entry 7). The desired product could still be formed in 42% yield under 10 bar of carbon monoxide (Table 1, entry 8). As the C2 feedstock, the reaction with ethylene was tested under different pressure, and excellent yields of the corresponding product was produced (Table 1, entries 9–11; for optimization details in Supplementary Information). It is also worthy to mention that the chirality of the obtained product was checked but no enantioselectivity was obtained. Only trace amount of the desired product could be detected when excess amount of aniline was added.

**Substrate scope**

After the initial optimization studies, we then investigated the generality of this cobalt-catalyzed carbonylation toward the

synthesize of γ-amino acid derivatives. As shown in Fig. 2, under our optimal conditions, a variety of phenols, alcohols, and amines that bearing a diversity of substituents were tested and gave good to excellent yields of the desired products in general. The corresponding products were obtained in moderate to good yields (**4 s, 4x, 5j, 5 v**) when using 5 equivalents of **1a**. Nucleophiles that bearing a heterocyclic moiety were all able to give the corresponding γ-amino acid derivatives in good yields (**4q, 4r, 4z, 5n, 5o, 5p**) with (R, R)-$^t$Bu-Pybox as the ligand. In addition, although the efficiency was better under 50 bar pressure, several examples were tested under lower pressure (5 bar ethylene and 5 bar CO) and the desired products were obtained in good yields (**4a, 4j, 5r, 5z**). Remarkably, a collection of functional groups, including halogens, ester, ketone, and nitrile, were all well adapted to this carbonylative transformation. However, no desired product could be detected when thiophenol or tert-butyl alcohol was applied as the nucleophile.

The generality of this one-pot procedure was tested by employing alkyl amines bearing various electron-withdrawing acyl groups (Fig. 3). In this reaction, acyl groups with different carbon chain degrees could all be used to give the target products with good yields (**6a, 6b, 6c**). Notably, when -CF$_3$ was employed in the reaction, the corresponding product was delivered in 51% yield (**6d**). Diethylacetamide and tetramethylurea were also applicable in this reaction system (**6e** and **6f**). For substrates containing multiple aminoalkyl sites, the carbonylation exclusively occurred on one site. For example, when using N-methyl pyrrolidone as a substrate, the carbonylation reacted at the C$_1$ site over the C$_2$ site, probably due to the stability of the C$_1$ radical (**6g**). On the

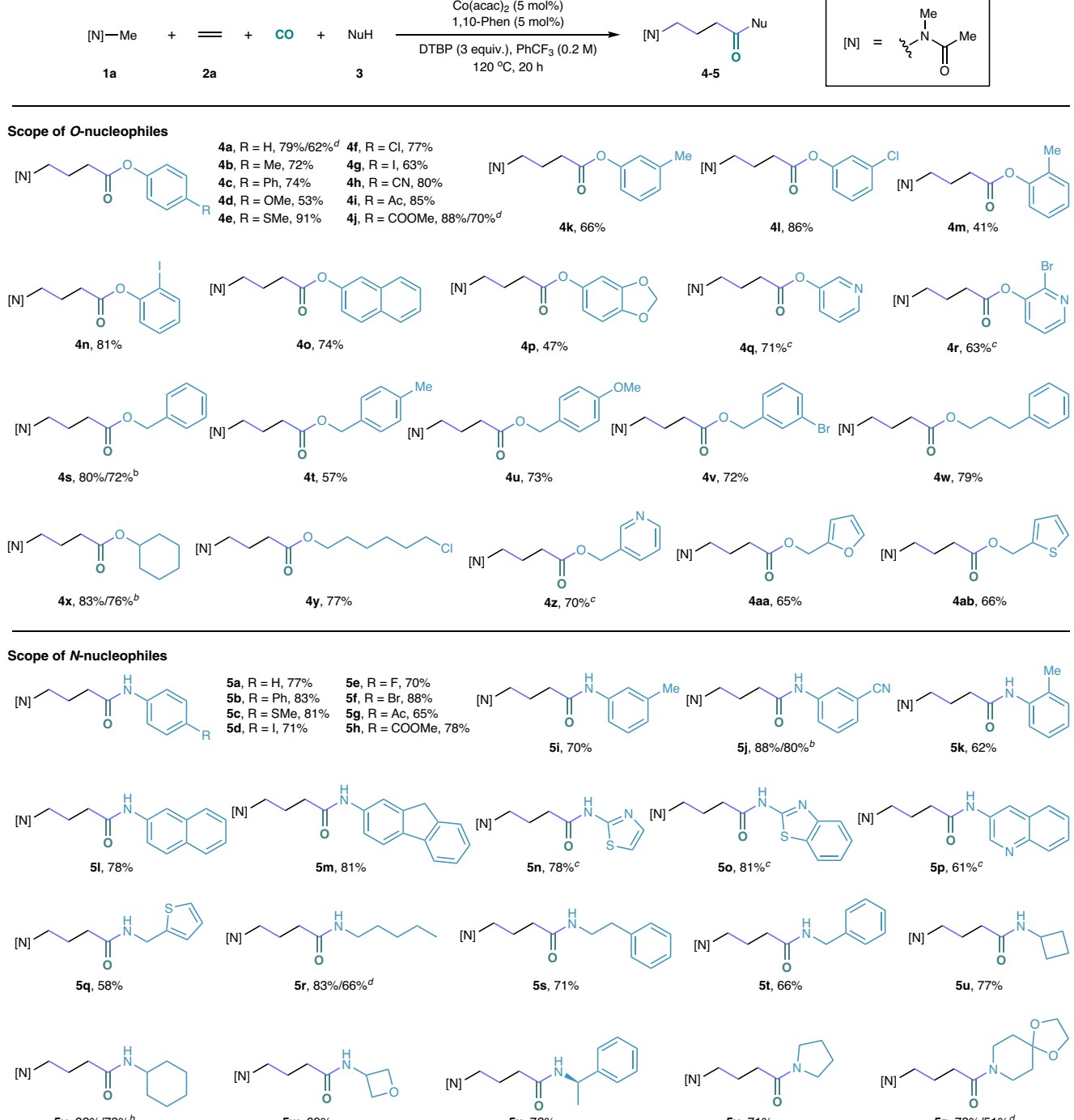

**Fig. 2 | Scope of the nucleophiles.** Reaction condition: [a]**1a** (10 equiv.), **3a** (0.3 mmol), Co(acac)$_2$ (5 mol%), 1,10-Phen (5 mol%), DTBP (3 equiv.), CO (40 bar), ethylene (10 bar), PhCF$_3$ (0.2 M), 120 °C, 20 h. [b]5 equiv. **1a**. [c](R,R)-$^t$Bu-Pybox (5 mol%). [d]5 bar ethylene, 5 bar CO.

contrary, the carbonylation occurred at the C$_1$ site over the C$_2$ site due to the sterically hindered C$_1$ site (**6i**). To our knowledge, only a few reports successfully conducted HAT process from the α-position of N-acylated primary amines, raising our concerns about this class of substrates[38,39]. To our delight, N-methylacetamide also proved to be suitable starting material for this transformation and been converted into the corresponding product in 57% yield by recrystallization (**6j**). In addition to the extensive application of nucleophilic partners and amide substrates, a variety of unactivated alkenes were also compatible with this transformation (Fig. 3). Various 2-hydroxyl substituted allylbenzenes were tolerated, delivering the target products by a

carbonylative cyclization (**7a-7e**). Next, substrates with a different group such as ester (**7k**), acyl (**7g**), halogen (**7p** and **7q**), and heterocyclic (**7u**) were all well-tolerated to deliver the corresponding γ-amino acid derivatives in good yields. In the cases of internal alkenes, no desired carbonylation product could be detected under our standard conditions.

As shown in Fig. 4, we then applied this procedure to the carbonylation of complex and biologically active molecules, which gave a series of γ-amino acid derivatives in good to excellent yields without further optimizations. This method for complex molecule compatibility is of great significance for drug discovery and

**Fig. 3 | Scope of amides and unactivated alkenes.** Reaction condition: [a]**1** (10 equiv.), ethylene (10 bar), **3** (0.3 mmol), Co(acac)$_2$ (5 mol%), **L3** (5 mol%), DTBP (3 equiv.), CO (40 bar), PhCF$_3$ (0.2 M), 120 °C, 20 h. [b]140 °C, 10 mol% Co(acac)$_2$, 10 mol% **L3**. [c]**2** (0.6 mmol), (R, R)-[t]Bu-Pybox. (5 mol%).

modifications. The reliability of this cobalt-catalyzed cascade carbonylation was further proven by synthesizing a series of dipeptide derivatives (Fig. 4). The dipeptide compounds, synthesized from various amino acid derivatives, such as D-phenylalanine methyl ester, D-tryptophan methyl ester, and N-Boc-4-amino-L-phenylalanine ethyl ester, successfully delivered in 67–78% yields (**9a**-**9c**). In addition, N-methylacetamide also demonstrated the efficiency of this reaction, furnishing the corresponding products in moderate yields (**9e**-**9g**).

## Synthetic applications
The practicality of this radical relay carbonylation reaction was presented by scale-up reaction and application to deliver 4-aminobutyric acid (GABA). Two scale-up transformations with unactivated alkene and ethylene gave similar yields (**5j**, 73%; **7j**, 62%) compared with the model reactions, respectively. Notably, the carbonylative cyclization product **10** was obtained in 50% yield when using amides containing alkenes. Six-membered rings are preferentially formed over five-membered rings due to the least

**Fig. 4 | Scope of complex molecules and peptides.** Reaction condition: [a]**1a** (10 equiv.), ethylene (10 bar), **3** (0.3 mmol), Co(acac)$_2$ (5 mol%), 1,10-Phen (5 mol%), DTBP (3 equiv.), CO (40 bar), PhCF$_3$ (0.2 M), 120 °C, 20 h . [b]Alkene (0.6 mmol), (R, R)-[t]Bu-Pybox (5 mol%). [c]140 °C. [d]Recrystallization.

repulsive forces between the chemical bonds of the six-membered ring product. In addition, the carbonylated products could be efficiently transformed into the corresponding γ-amino acids (**11** and **12**). Deacetylation to get **12** needs to be carried out under Schwartz reagent. Finally, diol could be successfully converted to the corresponding diesterification at both ends to form diamino acids derivatives (**13**, 41%).

## Mechanistic investigations

To gain insight into the reaction mechanism, a battery of control experimental studies was carried out (Fig. 5B). The addition of a radical scavenger to the reaction completely suppressed the formation of the desired product **7h** under otherwise standard conditions. In addition, the radical-trapping product **14** was detected by gas chromatograph-mass spectrometer, and radical-trapping

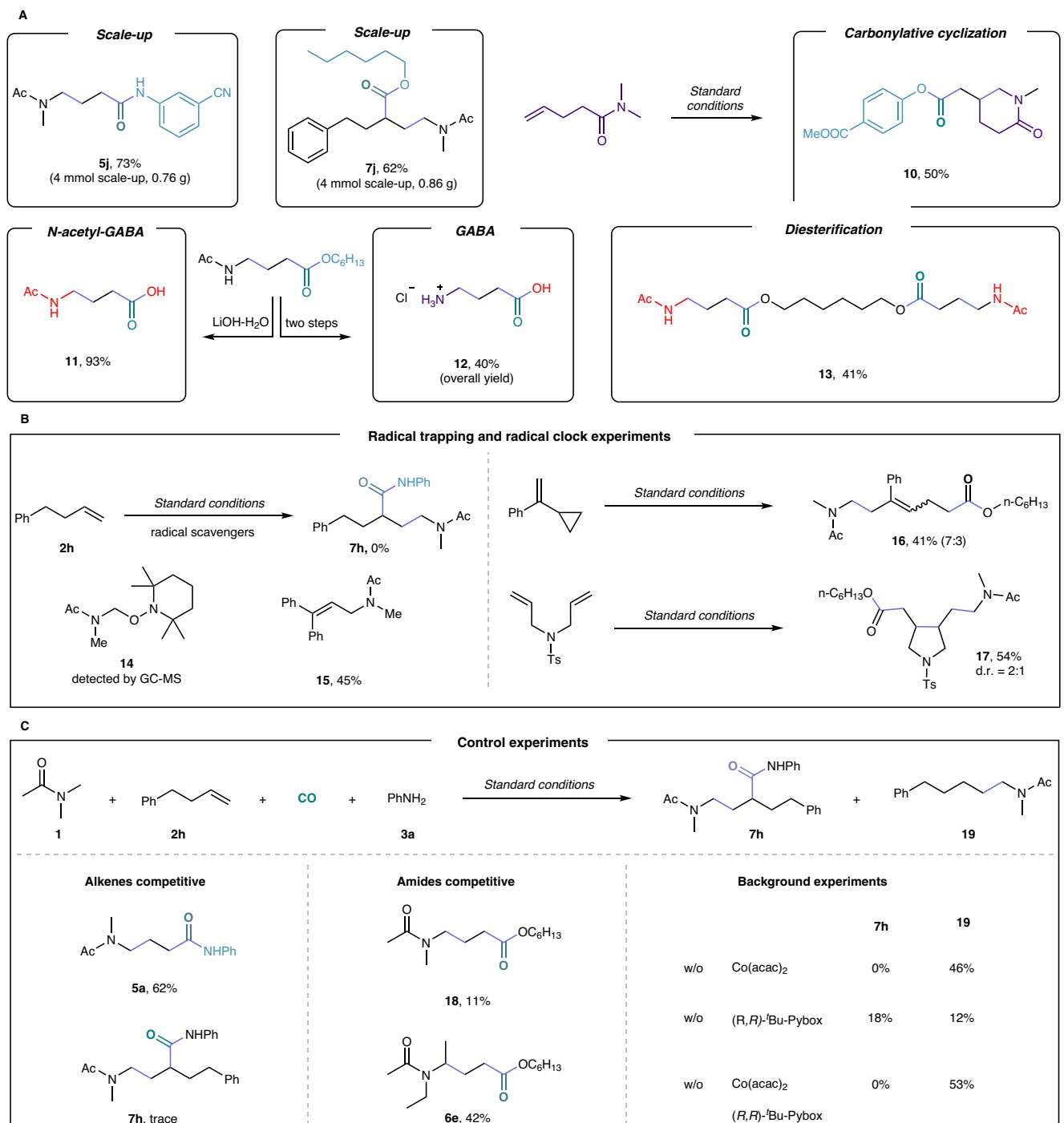

**Fig. 5 | Synthetic applications and mechanistic investigations. A** Scale-up experiments and synthetic applications. **B** Radical trapping and radical clock experiments.
**C** Control experiments.

product **15** was isolated in 45% yield. These results suggest that α-aminoalkyl radical may be involved in this process. To further confirm this fact, we conducted free radical clock experiments and obtained the corresponding ring-opening product **16** in 41% yield and ring-closing product **17** in 54% yield. Alkenes competition experiments showed that ethylene was more reactive than unactivated alkenes (Fig. 5C). Amides competition experiments turned out that the reactivity of -CH₂- is preferred to that of -CH₃, due to the stability of α-aminoalkyl radical. Finally, we performed this reaction without a catalyst and the direct hydroaminoalkylation product **19** was obtained in moderate yield.

Based on these mechanistic studies and previous reports on radical mediated reactions[40–43], we proposed a possible pathway for this radical relay carbonylation of alkenes and aminoalkyl radical (Fig. 6). Initial pyrolysis or single-electron oxidation to generate a tert-butoxyl radical, which could undergo a hydrogen atom transfer process with electron-withdrawing alkylamines **I** to generate an α-aminoalkyl radical intermediate **C**. Next, the resulting α-aminoalkyl radical **C** preferentially added to the terminal end of the alkene, delivering a new carbon radical species **D**. In path a, the radical species **D** was captured by the [Co¹] to generate metal intermediates **A**, and then carbon monoxide entered to obtain acyl metal **B**. In another path b, the

**Fig. 6 | Proposed mechanism.** A possible reaction pathway.

radical species **D** first captured carbon monoxide to give an acyl radical species **E**, which was subsequently captured by the metal. The final reductive elimination furnished the desired γ-amino acid derivatives and regenerated cobalt (I) for the next catalytic cycle.

In conclusion, we have developed a cobalt-catalyzed radical relay carbonylation of alkenes with various readily available building blocks. The desired γ-amino acid derivatives and peptides were produced in high yields in general with promising functional group tolerance. The work described herein not only solves a long-standing challenge in α-aminoalkyl radical addition to alkenes but also provides a strategy for the synthesis of γ-amino acid derivatives. This procedure features the direct use of abundant amine compounds as substrates, and substrates that range from the simplest ethylene to complex unactivated alkenes could all participate in the reaction. We believe that this work will inspire further exploration in the synthesis of complex amino acid molecules.

## Methods
### General procedure for the carbonylation of alkenes
A 4 mL screw-cap vial was charged with Co(acac)₂ (3.9 mg, 0.015 mmol, 5 mol%), (R, R)-ᵗBu-Pybox (4.9 mg, 0.015 mmol, 5 mol %). The vial was closed with a Teflon septum and cap and connected to the atmosphere via a needle. After amides (5-10 equiv.), PhCF₃ (1.5 mL, 0.2 M), nucleophile (1 equiv., 0.3 mmol), DTBP (131.6 mg, 0.9 mmol, 3 equiv.), and alkenes (0.6 mmol, 2 equiv.) (or ethylene, same as charging carbon monoxide) were added with a syringe under nitrogen atmosphere, the vial was moved to an alloy plate and put into a Parr 4560 series autoclave (300 mL) under an argon atmosphere. At room temperature, the autoclave was flushed with CO three times, then charged with 40 bar of CO. The autoclave was placed on a heating plate equipped with a magnetic stirrer and an aluminum block. The reaction mixture was heated to 120 °C for 20 h. After the reaction was complete, the autoclave was cooled down to room temperature and the pressure was released carefully. After cooling to room temperature, the reaction mixture was directly purified by column chromatography on silica gel using petroleum ether, ethyl acetate, and methanol to afford the corresponding product. *Note:* Because of the high toxicity of carbon monoxide, all the reactions should be performed in an autoclave. The laboratory should be well-equipped with a CO detector and alarm system.

## Data availability
The data supporting the findings of this study are available within the paper and its Supplementary Information or from the authors upon request.

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

## Acknowledgements

We thank the financial support from Chinese Academy of Sciences Dalian Institute of Chemical Physics (DICP) and K.C. Wong Education Foundation (GJTD-2020-08). Yang Yuan and Youcan Zhang gratefully acknowledge funding from the China Postdoctoral Science Foundation. Open Access funding enabled and organized by Projekt DEAL.

## Author contributions

L.-C.W. designed and carried out most of the chemical reactions, and analysed the data. L.-C.W. Y.Y. and Y.Z. provided raw material support. X.-F.W. designed and supervised the project. X.-F.W. and L.-C.W. wrote the manuscript.

## Funding

## Competing interests

The authors declare no competing interests.
