## [Peer Review File · Nature Communications]

REVIEWER COMMENTS

Reviewer #1 (Remarks to the Author):

In this paper, Wu and co-workers developed a novel strategy for the synthesis of γ -amino acid derivatives and peptides by cobalt-catalyzed aminoalkylative carbonylation of alkenes. The key to the success of the challenging aminoalkylative carbonylation of alkenes is using an acyl to protect the alkylamine, and tune its electronic property that favors the addition of alkenes. The alkene substrate scope is generally high with the simplest ethylene and monosubstituted alkenes working well. Various amides were also tested with good yields, and high regio-selectivity in the case with several potential reacting sites. N-methylacetamide with an N-H group is also applicable in this reaction, producing products that could be used for further transformations after removing the acetyl-protecting group. The radical mechanism is also supported by the radical trapping and radical clock experiments. Overall, this is a novel and useful carbonylation reaction for γ -amino acid derivatives synthesis, and I recommend the paper for publication in this Journal after minor revisions.

1. Is there any enantioselectivity for the mono-substituted alkene substrates when chiral Pybox ligand was used? If the mechanism of the reaction is through path A in Fig. 8, a chiral ligand might potentially induce an asymmetric combination of the cobalt with the alkyl radical to form a chiral cobalt species, followed by a CO insertion with retention of the chirality to the product.
2. The reactions use 10 equiv. of amides. Does it need large excess of amide to keep the selectivity? What is the case if the nucleophiles are in excess?
3. The starting material of products 11 and 12 should be denoted in Fig. 7.
4. The wavy bond of compound 16 seems in the wrong position. Is there an E/Z mixture, and what's the ratio?
5. The radical clock experiment of cyclopropane (Fig. 7) indicates that 1,1-disubstituted alkenes might work in this reaction, but it's a special case with a ring opening to form a less steric alkyl radical. I'm interested to know whether normal disubstituted alkenes work in this reaction. A more stable and bulkier alkyl radical will be formed in this case.
6. There should be a diastereoselectivity of the cyclic product 17, not just conformational isomers as described in the supporting information, page 78.
7. There is a 4.1 ppm 't' peak in several ¹H NMR spectra of compounds 7h, 7i, 7n, 7p, 7q, 7t. What's that? Is it an inseparable by-product?
8. The spectra width of most ¹H NMR is recommended to adjust by removing some blank low-field areas, which will keep the peaks clearer.

9. “ α -amino acids (11 and 12)” (below the Fig. 6), α should be “ γ ”.

Reviewer #2 (Remarks to the Author):

The submitted manuscript presents a study on the synthesis of γ -amino acid derivatives and peptides, specifically the cobalt-catalyzed aminoalkylative carbonylation of alkenes using DTBP as an additional oxidant. The authors have invested considerable effort in exploring reaction conditions, expanding substrate scopes, and investigating mechanistic aspects, which is duly acknowledged. However, after a thorough evaluation, it is with regret that I must recommend against the publication of this manuscript in Nature Communications.

Several substantial issues have been identified that render the manuscript unsuitable for inclusion in this journal. The novelty and innovation of the methodology presented raise concerns. While the approach does offer potential for application, it falls short in demonstrating a significant leap in innovation, especially the harsh reaction conditions employed in this study, notably the use of strong oxidants and high-temperature conditions. These conditions are incongruent with the prevailing trends in modern synthetic chemistry. Another substantial limitation is the poor atom economy resulting from the conversion of products to γ -amino acids. This issue contradicts fundamental principles of efficient synthesis and sustainability. Moreover, the compatibility of the developed methodology with peptide modifications remains uncertain. The rigorous demands of peptide synthesis, coupled with the harsh reaction conditions employed in this study, raise doubts about the viability of this approach for such intricate transformations. Further exploration and analysis of this aspect are warranted.

Considering these critical concerns, this reviewer believes that this manuscript should not be published in Nature Communications.

Reviewer #3 (Remarks to the Author):

In this scholarly manuscript, the authors have proffered an exposition detailing a cobalt-catalyzed radical relay carbonylation of alkenes employing a diverse array of readily accessible chemical substrates. The research divulged herein not only proffers a resolution to a longstanding quandary concerning α -aminoalkyl radical additions to alkenes but also furnishes an unprecedented modality for the synthesis of γ -amino acid derivatives.

This work is deemed meritorious of publication in Nature Communications, contingent upon the authors' comprehensive responses to the ensuing inquiries:

Concerning the spectrum of nucleophilic substrates, may thiophenol, in addition to alcohols, phenols, and amines, partake in the reaction?

Within the purview of nucleophilic substrates encompassing alcohols, can tertiary alcohols engage in the reaction?

Within the context of the experimental conditions table, what accounts for the substantial disparity in yield observed at 100°C versus 120°C?

In the mechanistic elucidation, subsequent to the generation of intermediate E, does the said intermediate merely undergo sequestration by the metal, or does it also engage in an additional reaction with ethylene?

REVIEWER COMMENTS

Reviewer #1 (Remarks to the Author):

In this paper, Wu and co-workers developed a novel strategy for the synthesis of γ -amino acid derivatives and peptides by cobalt-catalyzed aminoalkylative carbonylation of alkenes. The key to the success of the challenging aminoalkylative carbonylation of alkenes is using an acyl to protect the alkylamine, and tune its electronic property that favors the addition of alkenes. The alkene substrate scope is generally high with the simplest ethylene and monosubstituted alkenes working well. Various amides were also tested with good yields, and high regio-selectivity in the case with several potential reacting sites. N-methylacetamide with an N-H group is also applicable in this reaction, producing products that could be used for further transformations after removing the acetyl-protecting group. The radical mechanism is also supported by the radical trapping and radical clock experiments. Overall, this is a novel and useful carbonylation reaction for γ -amino acid derivatives synthesis, and I recommend the paper for publication in this Journal after minor revisions.

Thanks for your kind and valuable comments! Thank you!

1. Is there any enantioselectivity for the mono-substituted alkene substrates when chiral Pybox ligand was used? If the mechanism of the reaction is through path A in Fig. 8, a chiral ligand might potentially induce an asymmetric combination of the cobalt with the alkyl radical to form a chiral cobalt species, followed by a CO insertion with retention of the chirality to the product.

Thanks for your professional question! We had a similar question during this work. We tested ee, but no enantioselectivity was observed. That's also the reason we put two pathways in the proposed mechanism. A statement has been added in the main text.

2. The reactions use 10 equiv. of amides. Does it need large excess of amide to keep the selectivity? What is the case if the nucleophiles are in excess?

Thanks! The reaction starts from amide side and excess of amide to keep the reaction to proceed. If using excess of nucleophile, only trace amount of desired product could be detected. A statement has been added in the main text.

3. The starting material of products 11 and 12 should be denoted in Fig. 7.

Thanks! Added as suggested.

4. The wavy bond of compound 16 seems in the wrong position. Is there an E/Z mixture, and what's the ratio?

Thanks! It has been corrected and the ratio is 7:3. It has been added in Fig 7.

5. The radical clock experiment of cyclopropane (Fig. 7) indicates that 1,1-disubstituted alkenes might work in this reaction, but it's a special case with a ring opening to form a less steric alkyl radical. I'm interested to know whether normal disubstituted alkenes work in this reaction. A more stable and bulkier alkyl radical will be formed in this case.

Thanks! Disubstituted alkenes not working in this system. As you mentioned, bulkier alkyl radical is not suitable for this reaction which explains why disubstituted alkenes not working here. A statement has been added in the main text.

6. There should be a diastereoselectivity of the cyclic product 17, not just conformational isomers as described in the supporting information, page 78.

Thanks! The ratio is 2:1. It has been added in Fig 7.

7. There is a 4.1 ppm 't' peak in several ¹H NMR spectra of compounds 7h, 7i, 7n, 7p, 7q, 7t. What's that? Is it an inseparable by-product?

Thanks for your professional question! The signal is belong to the H close to N and been split due to amide group. The ¹H NMR has been corrected. Thanks!

8. The spectra width of most ¹H NMR is recommended to adjust by removing some blank low-field areas, which will keep the peaks clearer.

Thanks! All the NMR spectra have been corrected as suggested.

9. "α-amino acids (11 and 12)" (below the Fig. 6), α should be "γ".

Thanks! It has been corrected!

Reviewer #2 (Remarks to the Author):

The submitted manuscript presents a study on the synthesis of γ-amino acid derivatives and peptides, specifically the cobalt-catalyzed aminoalkylative carbonylation of alkenes using DTBP as an additional oxidant. The authors have invested considerable effort in exploring reaction conditions, expanding substrate scopes, and investigating mechanistic aspects, which is duly acknowledged. However, after a thorough evaluation, it is with regret that I must recommend against the publication of this manuscript in Nature Communications.

Several substantial issues have been identified that render the manuscript unsuitable for inclusion in this journal. The novelty and innovation of the methodology presented raise concerns. While the approach does offer potential for application, it falls short in demonstrating a significant leap in innovation, especially the harsh reaction conditions employed in this study, notably the use of strong oxidants and high-temperature conditions. These conditions are incongruent with the prevailing trends in modern synthetic chemistry.

Another substantial limitation is the poor atom economy resulting from the conversion of products to γ -amino acids. This issue contradicts fundamental principles of efficient synthesis and sustainability. Moreover, the compatibility of the developed methodology with peptide modifications remains uncertain. The rigorous demands of peptide synthesis, coupled with the harsh reaction conditions employed in this study, raise doubts about the viability of this approach for such intricate transformations. Further exploration and analysis of this aspect are warranted.

Considering these critical concerns, this reviewer believes that this manuscript should not be published in Nature Communications.

Thanks for your comments in any case! We have to emphasize that this is the first report on carbonylative synthesis of γ -amino acid derivatives.

Reviewer #3 (Remarks to the Author):

In this scholarly manuscript, the authors have proffered an exposition detailing a cobalt-catalyzed radical relay carbonylation of alkenes employing a diverse array of readily accessible chemical substrates. The research divulged herein not only proffers a resolution to a longstanding quandary concerning α -aminoalkyl radical additions to alkenes but also furnishes an unprecedented modality for the synthesis of γ -amino acid derivatives.

This work is deemed meritorious of publication in Nature Communications, contingent upon the authors' comprehensive responses to the ensuing inquiries:

Thanks for your kind and valuable comments! Thank you!

Concerning the spectrum of nucleophilic substrates, may thiophenol, in addition to alcohols, phenols, and amines, partake in the reaction?

Thanks! Thiophenol is not working in this system, only thioether was detected. A statement has been added.

Within the purview of nucleophilic substrates encompassing alcohols, can tertiary alcohols engage in the reaction?

Thanks! tert-butyl alcohol was tested but no desired product could be produced. A statement has been added.

Within the context of the experimental conditions table, what accounts for the substantial disparity in yield observed at 100°C versus 120°C?

Thanks! The activation of DTBP needs certain temperature. The low yield at 100 oC due to the low conversion of substrate. A statement has been added.

In the mechanistic elucidation, subsequent to the generation of intermediate E, does the said intermediate merely undergo sequestration by the metal, or does it also engage in an

additional reaction with ethylene?

Thanks! From literature, both are possible. However, we did not observe product from intermediate E addition with ethylene.

REVIEWERS' COMMENTS

Reviewer #1 (Remarks to the Author):

The authors have addressed all of the comments and questions that I raised in my first review.